# Human Histological Analysis of Early Bone Response to Immediately Loaded Narrow Dental Implants with Biphasic Calcium Phosphate^®^ Grid-Blasted Surface Treatment: A Case Report

**DOI:** 10.3390/dj11070177

**Published:** 2023-07-19

**Authors:** Tárcio Hiroshi Ishimine Skiba, Eduardo C. Kalil, Adriano Piattelli, Jamil Awad Shibli

**Affiliations:** 1Department of Periodontology, Dental Research Division, Guarulhos University, Guarulhos 07023-040, SP, Brazil; 2School of Dentistry, Saint Camillus International University of Health and Medical Sciences, Via di Sant’Alessandro 8, 00131 Rome, Italy

**Keywords:** dental implants, implant surface topography, osseointegration, human histology

## Abstract

Implant surface topography using bioactive material provides faster bone-to-implant healing. This histological report described the analysis of human bone tissue around an immediately loaded implant, with BPC^®^ (Biphasic Calcium Phosphate) grit-blasted surface treatment, after two months of healing. Two temporary mini-implants (2.8 × 10 mm) with BPC^®^ grit-blasting surfaces were placed and immediately loaded to retain a complete interim denture. After a 60-day healing period, one mini-implant was removed for histologic analysis. The ground section showed the whole implant surrounded by healthy peri-implant tissues. Implant surface presented a close contact with newly formed bone, showing some areas of osteoblasts secreting mineral matrix. The ground section depicted a bone contact of 60.3 + 8.5%. The BPC^®^ grit-blasted surface was biocompatible and enabled the osseointegration process after a short-term period.

## 1. Introduction

Only after surgical installation does the osseointegration process begin due to the biocompatibility of the implant surface topography. The capacity of a material to elicit the optimal biological response for a particular scenario or application is known as biocompatibility [1,2]. This property enhances the characteristics of the implant surface topography and the bone tissue healing around the implant surface. 

A set of biological processes are started after the installation of dental implants to aid in the healing of the peri-implant wound. The initial matrix formation required to enable successful osseointegration is provided by the recruitment and migration of osteogenic cells to dental implant surface topography. As a result, the implant surface topography is crucial to this process, especially in the initial stages of recovery. Additionally, various molecular and cellular mechanisms of the initial bone matrix production and mineralization process are affected by the surface topographical characteristics at the micro- and nanoscale [3,4]. 

Long-term implant longevity highly depends on the amount of mineralized new bone tissue that is in direct contact with the implant surface topography. Long-term implant success depends on a number of factors, including implant material, bone quality and quantity, surgical technique, surface characteristics, implant design, and implant loading scenarios.

Numerous advancements have taken place in the last ten years to increase bone-to-implant contact (for a review, see [3]). To increase the roughness of the implant surface, for instance, the industry has employed a number of techniques, including titanium anodization, discontinuous calcium phosphate crystal deposition, coatings with biological molecules, and chemical modification techniques, including sandblasting, acid-etching, and grit-blasting [3,4,5,6,7].

In the past, machined surfaces, commonly referred to as “turned” or “smooth,” were most frequently employed. The main feature of the machined surfaces was that distance osteogenesis characterized the pattern of bone formation [3]. After that, surfaces that had been sandblasted and acid-etched were created using a combination technique that involved blasting (to generate a macro-texture) and acid etching (to create a final micro-texture). Surfaces that had been sandblasted and acid-etched exhibited strong osteoconductive qualities and the ability to promote cell growth. Additionally, “contact osteogenesis”—where the osteoblasts begin to deposit an osteoid matrix right on the implant surface—is a feature of the bone-growth pattern surrounding blasted, rough surfaces [3,4,5,6,7]. 

BPC^®^ (biphasic calcium phosphate) is a mix of hydroxyapatite and beta-tricalcium phosphate, natural elements of bone tissues, which have improved chemical characteristics that allow faster direct bone-to-implant contact [2]. Complementarily, BPC^®^ as a grit-blasting material can provide degrees of roughness comparable with those of other materials commonly used, such as Alumina-Ti. BPC^®^ attains roughness ranges of 1–2 μm, considered ideal for the adequate biochemical anchorage of dental implants [4]. Histological evaluations (preclinical and some case human reports) and clinical studies (implant stability after implant-supported restoration placement) have measured osseointegration [3]. Therefore, evaluation of the bone–implant interface is crucial to understanding the relationship between these methods of implant surface treatment and the osseointegration process. All these studies provide valuable data, but with limited or low evidence quality. It is important to point out that the results obtained from in vitro and animal studies could not be automatically translated to a human situation. Accordingly, it is very important to evaluate retrieved human implants. These retrieved implants can be removed due to a wide range of technical and biological problems. Technical problems include unrestorable prosthetics, misalignment, sub-optimal positions from aesthetic and hygienic points of view, and the inability of an implant to meet changed prosthetic needs. Biological problems, such as mobility, fracture, peri-implanitis, bone loss, and infection, can also result in implant removal. Implants from humans can also be obtained as part of a research protocol approved by an Ethical Committee using experimental implants with reduced dimensions (micro-implants) or temporary implants to support an interim restoration. All these methodologies aim to facilitate the retrieval process. In the aforementioned situations, the dental implants present bone tissue attached to the implant surface that allow a better understanding of the whole picture of the biological process of bone-to-implant contact. Understanding the failure mechanisms or the responses of the peri-implant tissues (soft tissues and bone) can also be greatly aided by the meticulous study of all these various types of implants [3,5,6].

Therefore, evaluating dental implants retrieved from human jaws in a non-diseased situation is a valuable source of information about the bone–implant interface. This case report aimed to assess the impact of BPC^®^ implant surface topography immediately loaded in the human maxilla and retrieved after 60 days of healing.

## 2. Materials and Methods

A 46-year-old female patient, whose complaint was the instability of a maxillary removable complete denture, presented at Guarulhos University Dental School for implant-supported rehabilitation treatment. After a comprehensive clinical and radiographic examination, which showed adequate bone volume, and after discarding any changes in any systemic disease, the placement of 6 dental implants to support a fixed prosthesis in the maxilla was planned.

Among the 6 implants placed in the patient were two narrow 2.8 mm wide and 10 mm long screw-shaped dental implants (Axiom^®^, Anthogyr, Salanches, France) with surfaces treated by BPC^®^ grit-blasting, and etched with nitric acid. These mini-implants were placed in the edentulous maxilla as “temporary” implants in addition to the six conventional implants (4.1 mm diameter and 10 to 13 mm length) according to manufacturer instructions (Figure 1). The temporary mini-implants were immediately loaded to retain a complete interim denture during the healing period of the standard implants (diameter ≥ 3.5 mm). After two months of healing without any post-operative complaints or complications (Figure 2), during the second-stage procedure to uncover the 6 conventional implants, the left maxillary temporary mini-implant and abutment were removed, together with the surrounding bone using a 4.0 mm internal diameter trephine burr (Figure 3). After removal, the specimen was fixed by immediate immersion in 4% phosphate-buffered formalin. The second temporary implant placed on the right side was removed using a retriever device. This report was approved by the Ethics Committee of the University of Guarulhos (UnG), Sao Paulo, Brazil (IRB # UnG 60830316.8.0000.5506).

### Processing of Specimens and Histometric Analysis

The specimen was preserved for ten days in 4% phosphate-buffered formalin at pH seven before being moved to a 70% ethanol solution to await processing. The specimen was hard-sectioned using the Donath–Breuner method [8], infiltrated and embedded in LR White resin (London Resin Company, Berkshire, UK), and then dehydrated in progressively higher alcohol concentrations up to 100%. A part that represented the site’s center had been prepared. The portion was micro-ground and polished until it had a thickness of around 25 mm. It was ready for histomorphometry after being stained for optic microscopic examination using Stevenel’s blue and alizarin red stain. The percentage of implant bone in direct contact with the implant surface topography over the whole implant’s perimeter is known as the percentage of bone-to-implant contact (%BIC). A light microscope (NIS, Nikon; Melville, NY, USA) that was attached to a high-resolution camera and interfaced with a display and personal computer was utilized for the assessment. This optical device was coupled with a digitizing pad and a historical software program from Nikon that included image-capture features.

## 3. Results

Cortical bone was seen in the first 1 to 3 coronal millimeters of the ground slice (Figure 4). There was mineralized, recently formed bone in close proximity to the implant surface at the interface of the retrieved implant, with no gaps or fibrous connective tissue. This newly produced bone conformed to the surface imperfections of the dental implant (Figure 5). At several contact sites, there were areas of bone remodeling with numerous reversal lines. The implant surface actually made up a portion of the osteocyte wall in some areas of the dental implant perimeter. There were large osteocyte lacunae in the developing trabecular bone. Osteoblasts were seen depositing unmineralized osteoid matrix onto the metal surface in some of the implant’s retrieved areas (Figure 6). With apposition and resorption processes, newly produced bone filled the bulk of the inter-thread voids. Epithelial down growth and inflammatory cell infiltration were not present. Bone-to-implant contact was 60.3 + 8.5 percent.

## 4. Discussion

The exact amount of bone needed at the interface of loaded implants to preserve long-term clinical success is still unknown [9,10]. Different amounts of BIC % were found in the literature, reporting results ranging from 18% to 65% [5,9,10,11,12,13]. The present study showed a high % of BIC (60.3 ± 8.5%) and the osseointegration process of peri-implant tissue on a temporary implant with a BPC^®^ grit-blasted surface, even at the early stage of tissue healing after immediate loading.

Among other factors such as surgical technique and patient variables (smoking and health conditions such as diabetes and osteoporosis), the implant surface topography plays a vital role in BIC%, since it will be an essential factor in the peri-implant cellular and molecular mechanisms [3].

Over the years, several changes have been made in dental implants’ macro- and micro design. These modifications aimed to improve the bone tissue healing around dental implants, allowing higher success rates in the immediate loading protocols, avoiding a second-stage procedure. To load implants immediately, implants with treated surfaces that can stimulate new bone formation and increase the value of BIC, reducing the healing time, represent an important tool from a clinical point of view [7,11,12].

The implant surface topography also improves the clinical outcomes of implant-supported restorations in areas of type IV bone [3,9,13]. In addition, studies have shown that roughened [14,15] and moderately roughened titanium surfaces can provide better osseointegration than smoother surfaces [15]. The implant surface treatment can be performed by several methods, such as additive processes, e.g., titanium plasma spraying and surface coating with biomimetic materials, or subtractive mechanisms, by grit-blasting and acid-etching [11,16,17,18,19]. One of the most common protocols for implant surface treatment is using grit-blasting followed by acid etching. The most common material used for grit-blasting is alumina (Al_2_O_3_), a low-biocompatibility material [17]. However, by being insoluble in acid, alumina gets partially trapped on the implant surface, which can compromise the implant’s osseointegration or even decrease titanium’s corrosion resistance in a physiological environment [17]. A possible alternative to alumina for roughening titanium dental implants would be to use a mix of calcium phosphates such as hydroxyapatite and beta-tricalcium phosphate (biphasic calcium phosphate). This leads to a biocompatible, osteoconductive, and resorbable material [9,16], entirely soluble in acid, reducing the residual material trapped on the roughened implant surface.

Furthermore, instead of being prejudicial to the osseointegration process, this minimum residual amount will optimize it [2]. The results from this study corroborate this concept, since it was possible to see new and woven bone formation with bone cells in direct contact with the grit-blasted BPC implant surface, even at an early stage of tissue healing after immediate loading. In addition, these findings agree with previous studies that showed that immediate loading allowed new bone formation around the implant surface in both the maxilla and mandible [3,7,13].

Peri-implant bone tissue undergoes remodeling, with a transformation of the initially produced woven bone into a bone with a lamellar configuration, showing a higher degree of organization. Earlier protocols suggested a submerged healing period of 4–6 months to obtain mineralized bone tissue at the implant surface interface, and an earlier implant loading could result in fibrous tissue formation at the bone–implant interface. Conversely, several researchers [20,21,22] have reported, in the last 2 decades, that in early and immediately loaded implants placed in good-quality bone, it was possible to obtain a high level of osseointegration, clinically, radiographically, and histologically similar to that of implants used with a standard submerged protocol [20]. This procedure, when correctly indicated, provided the resolution of several issues. Many patients complained about how painful it was to wear temporary prostheses; therefore, it would be advantageous for most of them if there was a way to speed up the healing process without compromising the long-term success of the dental implants. It was feasible to see peri-implant lamellar bone organized in haversian systems in our case report. Because the remodeling processes most likely started from the implant surface and moved outward, these systems near to the implant surface were mostly constructed in a parallel manner [3,21].

Complementary mechanical stimuli controlled cell division and differentiation, as well as the kind and structure of the tissue. The sighting of numerous osteocytes at the implant surface suggests the role of these cells as mechanosensory cells [22]. Additionally, assessing the healing rates at the interface at various time points could be beneficial. The fact that bone’s hardness and elastic modulus tend to rise with time may indicate that osseointegration is a highly dynamic process that involves the bone’s adaptation to functional loading stimuli to enhance the bone’s overall biomechanics [23].

We were able to assess the interaction between the implant surface and the peri-implant tissues by removing implants from individuals and retaining their bone anchoring by evaluating this tissue in undecalcified sections. In the case study, a temporary implant was employed to give the patient some stability and comfort from the entire provisional denture. This allowed the implant to be withdrawn safely from the patient without causing any harm. This made it possible to perform a histological analysis of the osseointegration and bone-healing processes around the dental implant. Although macro- and micro-topographies of implants have been examined in pre-clinical studies [3,13,23], suggesting that the implant surface characteristic may be able to modulate the bone response during the healing phase, not all of the evidence from these studies can be applied to a human situation [3]. As a result, information gathered from implants removed from human jaws continues to be a crucial source. Last but not least, randomized clinical trials are required to more fully comprehend the results of this sort of surface therapy in comparison to others.

## 5. Conclusions

Within the limits of this case report, it could be concluded that the BPC^®^ grit-blasted surface proved biocompatible and enabled the process of osseointegration in temporary implants with immediate loading.

## Figures and Tables

**Figure 1 dentistry-11-00177-f001:**
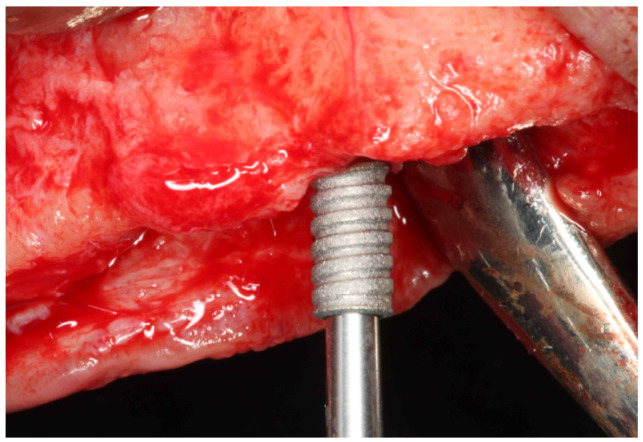
Clinical view of the temporary implant placement.

**Figure 2 dentistry-11-00177-f002:**
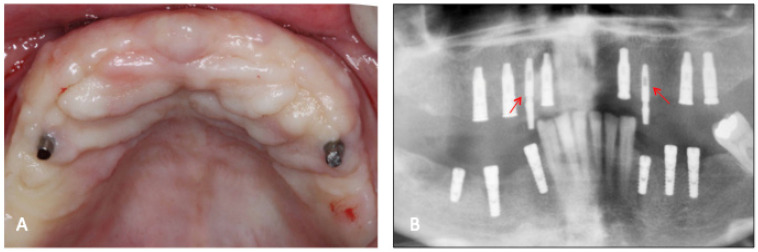
(**A**) Clinical aspect of the temporary implants after 60 days of healing; (**B**) radiographic aspect of the temporary implants (arrows) and the conventional type. Note that the conventional implants were submerged and will receive the 2nd stage of surgery.

**Figure 3 dentistry-11-00177-f003:**
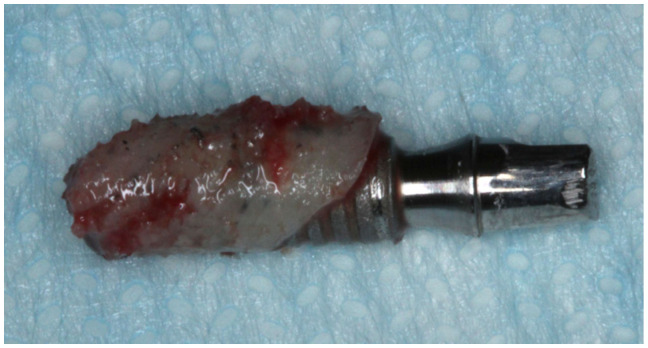
Temporary implant retrieved after 60-day healing period. Note the presence of attached bone tissue around the implant.

**Figure 4 dentistry-11-00177-f004:**
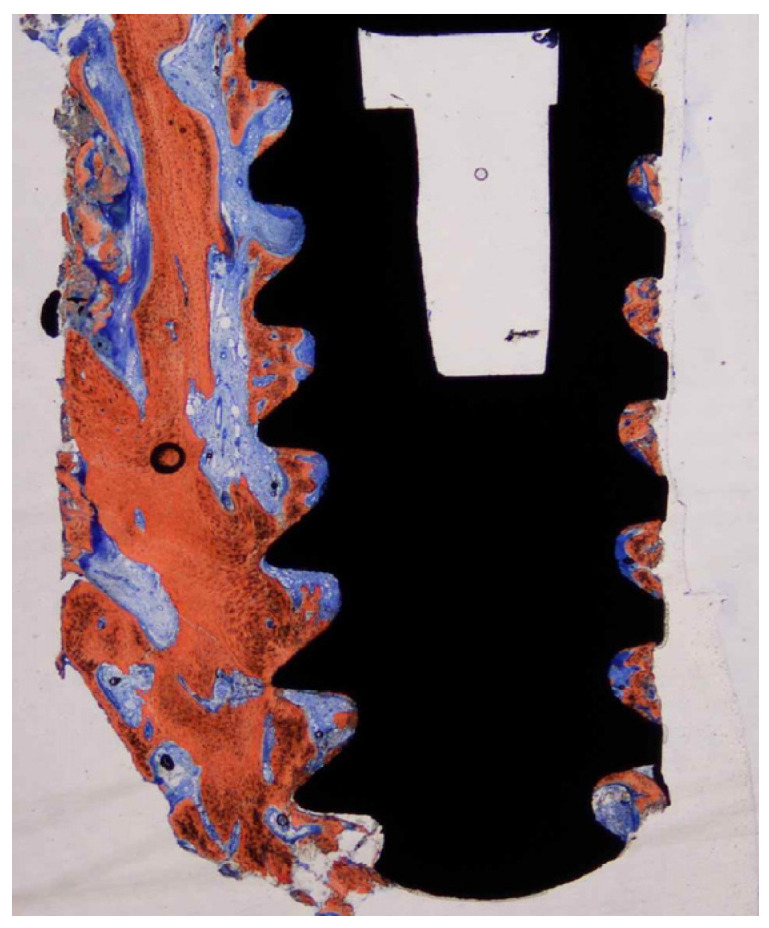
Ground section—15× original magnification; Stevenel’s blue and alizarin red stain. The implant is surrounded by newly formed bone along the entire perimeter with some areas of woven bone.

**Figure 5 dentistry-11-00177-f005:**
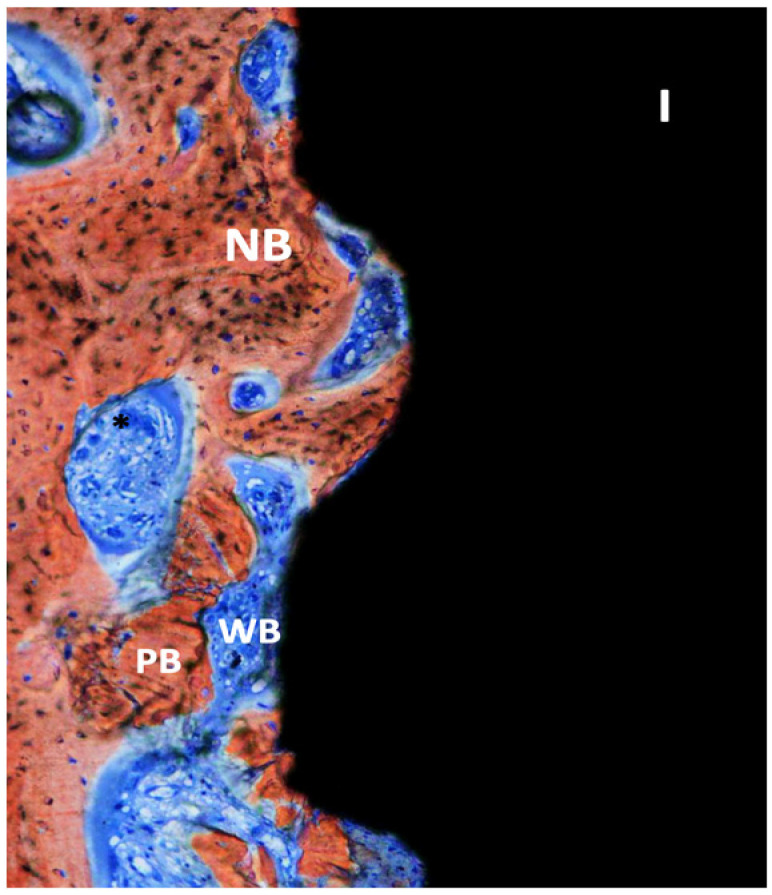
Ground section—200× original magnification; Stevenel’s blue and alizarin red stain. The implant (I) is surrounded by new bone (NB) with areas of ongoing bone formation. Woven bone (WB) and some areas with connective tissue are also seen with vessels (*) and some particles of pristine bone (PB).

**Figure 6 dentistry-11-00177-f006:**
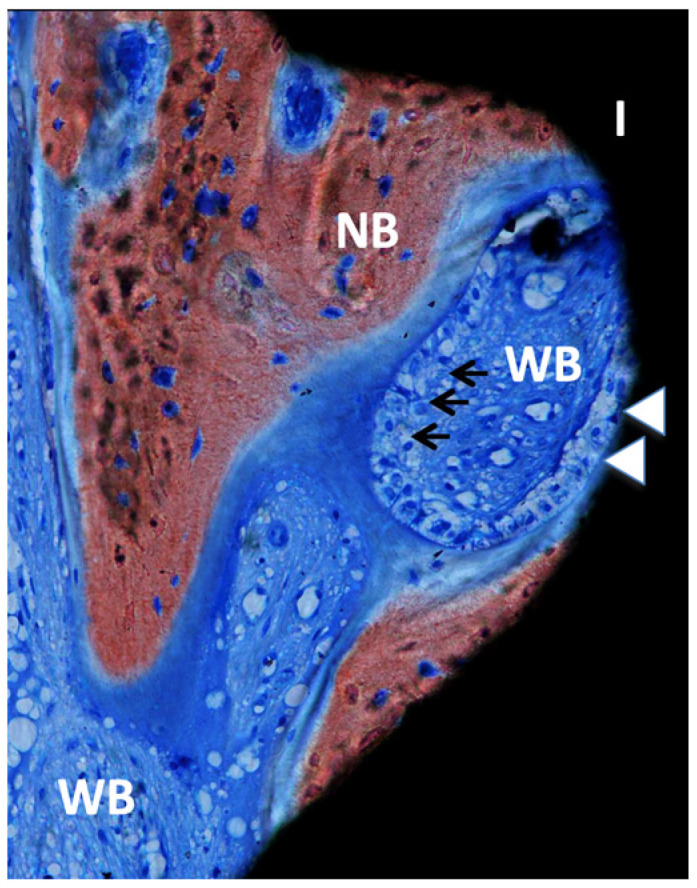
Ground section—200× original magnification; Stevenel’s blue and alizarin red stain. The implant (I) is in close contact with osteoblasts (white arrowheads), suggesting ongoing bone formation or osseointegration. New bone (NB) is also present with areas of bone matrix formation with several osteoblasts (arrows) and a large area of woven bone (WB), usually seen in the regions with type IV bone (maxillae).

## Data Availability

The data are unavailable due to privacy or ethical restrictions. Please contact the corresponding author.

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
