# Peer review of "Human Histological Analysis of Early Bone Response to Immediately Loaded Narrow Dental Implants with Biphasic Calcium Phosphate® Grid-Blasted Surface Treatment: A Case Report"

_dentistry, 2023, doi:10.3390/dj11070177_

Round 1

Reviewer 1 Report

This is a very interesting case report on the osseointegration of dental implants. Performing this kind of study with humans is challenging due to obvious reasons. Although there are some case reports of this type published in the past, I consider this paper as highly significant and original due to: (1) the use of a novel surface treatment; (2) paucity of this kind of data from humans.

I have some suggestions to improve this manuscript, which can be found in the attached file. I could find no major issue, with most suggestions referring to English editing.

As mentioned above, there are some minor issues to solve. However, it is a well written manuscript.

Author Response

June 29th, 2023.

To: Editors of Dentistry Journal

Ref.: Manuscript Dentistry -2334457

Human Histological Analysis of Early Bone Response to Immediately Loaded Narrow Dental Implants with BCP® Grid-Blasted Surface Treatment: a case report

Dear Editors

Thank you very much for considering the manuscript entitled above for publication in the Dentistry Journal.  I would like to thank the reviewers for their comments, which surely will improve this histological study. The revised manuscript was highlighted in yellow to facilitate the revision.  The English language was also revised by 2 of the authors (JAS and AP). Please, find below point-by-point amendments for each reviewer.

Warmest regards from Brazil,

Jamil (on behalf of all authors).

Reviewer #1

Concern of the reviewer: This is a very interesting case report on the osseointegration of dental implants. Performing this kind of study with humans is challenging due to obvious reasons. Although there are some case reports of this type published in the past, I consider this paper as highly significant and original due to: (1) the use of a novel surface treatment; (2) paucity of this kind of data from humans.I have some suggestions to improve this manuscript, which can be found in the attached file. I could find no major issue, with most suggestions referring to English editing.

Our response: Thanks for your comments and review. The manuscript was revised accordingly. Please, see the revised text.

Revised text: N.A

Reviewer 2 Report

Thank you for your interesting work. The present study evaluated the early bone response of immediate loaded implants that are treated with calcium biphosphate biomaterials in human situation. There are some minor points the authors should focus to: 

Introduction

The osseointegration is measured by histological (preclinical and some case human reports) and clinical studies (implant stability after implant-supported restoration placement) Therefore, evaluation of the bone-implant interface is crucial to understanding the relationship between these methods of implant surface treatment and the osseointegration process All these studies provide valuable data, however, with limited or low evidence quality” 

Comment [unclear sentence]

Materials and Methods

Detailed surgical procedure for implant placement is missing.

Surface characteristics of the tested implant is needed to address.

Discussion

Pls discuss about possible osteogenesis phenomenon based on the current histological results.

It’s better to discuss about the biocompatibility of tested implant at short-term period?

General comment

It is strongly recommended that the revised manuscript is reviewed to correct any grammatical or spelling errors.

Author Response

Reviewer #2

Concern of the reviewer: Thank you for your interesting work. The present study evaluated the early bone response of immediate loaded implants that are treated with calcium biphosphate biomaterials in human situation. There are some minor points the authors should focus to: 

Introduction

“The osseointegration is measured by histological (preclinical and some case human reports) and clinical studies (implant stability after implant-supported restoration placement) Therefore, evaluation of the bone-implant interface is crucial to understanding the relationship between these methods of implant surface treatment and the osseointegration process All these studies provide valuable data, however, with limited or low evidence quality” 

Comment [unclear sentence]

Our response: Thanks for your comment. The main idea of the sentence is to show that some preclinical studies are not able to translate the results to clinical situations.

Revised text: N.A

Concern of the reviewer: Materials and Methods: Detailed surgical procedure for implant placement is missing. Surface characteristics of the tested implant is needed to address.

Our response: Well pointed. We revised the manuscript accordingly.

Revised text: Among the 6 implants placed in the patient were two narrow 2.8mm wide and 10mm long screw-shaped dental implants (Axiom®, Anthogyr, France) with surfaces treated by BPC® grit-Blasting, and etched with nitric acid. These mini-implants were placed in the edentulous maxilla as “temporary” implants in addition to the six conventional implants (4.1mm diameter and 10 to 13mm length) according to manufacturer instructions (Fig. 1).

Concern of the reviewer: DiscussionPls discuss about possible osteogenesis phenomenon based on the current histological results. It’s better to discuss about the biocompatibility of tested implant at short-term period?

Our response: Thanks for the comment. The biocompatibility of the implant (and all implantable material in the human body) is tested using ISO 10993-1. This ISO states that all materials must be tested for biocompatibility in animal tests (mostly rabbit tibia, in the case of dental implants and biomaterials). That’s the reason we decide to discuss the osseointegration process.

Revised text: N.A

Reviewer 3 Report

Dear editor and authors, 

This is a well-prepared case report which indicates the BPC® grit-blasted surface was biocompatible and enabled the osseointegration process after a short-term period.  It presents a well-delineated method and adequate processing of the data for a definite outcome. Therefore, my decision is acceptance.

Dear editor and authors, 

This is a well-prepared case report which indicates the BPC® grit-blasted surface was biocompatible and enabled the osseointegration process after a short-term period.  It presents a well-delineated method and adequate processing of the data for a definite outcome. Therefore, my decision is acceptance.

Author Response

Reviewer #3

Concern of the reviewer: Dear editor and authors, This is a well-prepared case report which indicates the BPC® grit-blasted surface was biocompatible and enabled the osseointegration process after a short-term period.  It presents a well-delineated method and adequate processing of the data for a definite outcome. Therefore, my decision is acceptance.

Our response: Thank you for the comment.

Revised text: N.A

Reviewer 4 Report

The paper under revision is a human histological analysis of an immediately loaded narrow implant with a new surface (BCP Grid Blasted). The content is original and innovative especially from the histological perspective not very common in implant clinical dentistry. However, being a case report with ethical and consent implications due to data processing, it would be essential to show that a proper consent taking was performed. Additionally a bit of patient's history and clinical information would be essential to understand why the treatment planning has been made this way and if there was a proper risk assessment carried out before the procedure. Being nowadays the implant procedures carried out with a sound and extensive pre-op documentation, I couldn't find any track of CBCT scans nor any image regarding restorative planning ahead of the treatment itself. Although this missing clinical documentation, the scientific contribution is remarkable in terms of the histological analysis and the outcomes. My request is to proceed with minor revisions to implement the missing parts.

Minor English proofreading could be advisable.

Author Response

Reviewer #4

Concern of the reviewer: The paper under revision is a human histological analysis of an immediately loaded narrow implant with a new surface (BCP Grid Blasted). The content is original and innovative especially from the histological perspective not very common in implant clinical dentistry. However, being a case report with ethical and consent implications due to data processing, it would be essential to show that a proper consent taking was performed. Additionally a bit of patient's history and clinical information would be essential to understand why the treatment planning has been made this way and if there was a proper risk assessment carried out before the procedure. Being nowadays the implant procedures carried out with a sound and extensive pre-op documentation, I couldn't find any track of CBCT scans nor any image regarding restorative planning ahead of the treatment itself. Although this missing clinical documentation, the scientific contribution is remarkable in terms of the histological analysis and the outcomes. My request is to proceed with minor revisions to implement the missing parts.

Our response: Well pointed. As discussed in the M&M, the case was properly evaluated and the IRB consent was signed by the patient.

Revised text: N.A

Reviewer 5 Report

Dear Authors,

In file you will receive my comments and suggestions.

All the best!

Author Response

Reviewer #5

Dear authors,

The topic covered for this Case Report Article is of major one and current interest.

The surface of dental implants and the Osseo integrative response at the level of the human

body is a challenge for dental clinical field, bioengineering and material engineering at the same

time.

I would like to congratulate you on the Manuscript and at the same time I would like to have

some questions and remarks on the substantive part of the theme.

Concern of the reviewer: 1. What is your experience with the Bone Level, TissueLevel and Narrow implants, so that you chose to do the research on the Axiom BL2.8 implants produced by Anthogyr? Detailing this aspect would bring a scientific layer to the Manuscript.

Our response: Thanks for the comments. There was no specific reason for the choice of the implant, except that at the time, just a few brands presented narrow implants with 2-pieces (implant and abutment).

Revised text: N.A

Concern of the reviewer: 2. In the Manuscript, it is not clearly indicated whether the BCP grit-blasted method facilitates an optimal roughness of the implant surface for the immediate initiation of the osseointegration process, or is it a method that produces a surface free of contaminants that can appear over the classic sandblasting method?

Our response: Well pointed. This report did not evaluate specifically the surface process method itself, but the behavior of a specific implant surface. This issue was pointed out in the discussion section.

Revised text: The implant surface topography also improves the clinical outcomes of implant-supported restorations in areas of type IV bone [3,9,13]. In addition, studies have shown that roughened [14,15] and moderately roughened titanium surfaces can provide better osseointegration than smoother surfaces [15]. The implant surface treatment can be performed by several methods, such as additive processes, e.g., titanium plasma spraying and surface coating with biomimetic materials, or subtractive mechanisms, by grit-blasting and acid-etching [11,16-19]. One of the most common protocols for implant surface treatment is using grit-blasting followed by acid etching. The most common material used for grit-blasting is alumina (Al2 O3), a low biocompatibility material [17]. However, by being insoluble in acid, alumina gets partially trapped on the implant surface, which can compromise the implant's osseointegration or even decrease titanium's corrosion resistance in a physiological environment [17]. A possible alternative to alumina for roughening titanium dental implants would be to use a mix of calcium phosphates such as hydroxyapatite and beta-tricalcium phosphate (Biphasic calcium phosphate). This leads to a biocompatible, osteoconductive, and resorbable material [9,16], entirely soluble in acid, reducing the residual material trapped on the roughened implant surface.

Concern of the reviewer: 3. Axiome 2.8/10mm is a narrow-type implant for small mesio-distal spaces, even called an implant for the lateral incisor, but of course it is also used as a temporary implant to support temporary removable prostheses until the solution of a fixed screwed oral rehabilitation is available. The manuscript mentions the immediate loading by supporting the transitory full denture, but in Figure 2a, a prosthetic abutment for fixed and luting prosthetic is seen at the supra-gingival level and not a specific attachment system (ball or locator) of a removable prosthetic work. In this case, do you consider that the loading is specific to the selected implant and if its connection with the prosthetic part does not influence the final result BIC%?

Our response: Well pointed. The present report shows the off-label use of narrow implants for research purposes only. The manufacturer did not recommend the use of narrow implants for overdentures (they neither have oring or locator abutments for narrow implants). Finally, the BIC results found in this report were influenced by all factors pointed out by you. Thank you for bringing up some important points for the discussion.

Revised text: N.A

Reviewer 6 Report

Case report title: Human Histological Analysis of Early Bone Response to Immediately Loaded Narrow Dental Implants with BCP® Grid- 3 Blasted Surface Treatment: a case report.

This report is well presented, and clinicians have explained each step of implant placement and bone regeneration clearly.

Abstract: Well written and within the word limits. Keywords: simple and understandable.

Introduction: Well-written and explained.

Material and method: Well explained

Result: Well explained

Discussion: Well-written and within the scope of the report.

Conclusion: Well-written and concise.

References are properly marked and no duplication is seen.

Good

Author Response

Reviewer #6

Concern of the reviewer: his report is well presented, and clinicians have explained each step of implant placement and bone regeneration clearly.Abstract: Well written and within the word limits. Keywords: simple and understandable; Introduction: Well-written and explained; Material and method: Well explained; Result: Well explained; Discussion: Well-written and within the scope of the report; Conclusion: Well-written and concise.

References are properly marked, and no duplication is seen.

Our response: Thanks for your comments.

Revised text: N.A

Round 2

Reviewer 2 Report

The quality of present manuscript (dentistry-2334457.R1) has been improved according to the suggestions of reviewers. It is suitable for publication in this journal.

Reviewer 4 Report

N/a

Reviewer 5 Report

Good luck in future research